# Analysis of microRNA Expression Profiles in Broiler Muscle Tissues by Feeding Different Levels of Guanidinoacetic Acid

Mengqian Liu [1,†], Mengyuan Li [1,†], Jinrui Ruan [1], Junjing Jia [1,2], Changrong Ge [1,2,*] and Weina Cao [1,2,*]

[1] College of Animal Science and Technology, Yunnan Agricultural University, Kunming 650201, China; lmq7498a123@163.com (M.L.); lmy15099996948@163.com (M.L.); rjrz2928@163.com (J.R.)
[2] Yunnan Provincial Key Laboratory of Animal Nutrition and Feed, Yunnan Agricultural University, Kunming 650201, China
[*] Correspondence: gcrzal@126.com (C.G.); caoweina@ynau.edu.cn (W.C.)
[†] These authors contributed equally to this work.

**Abstract:** The aim of this study was to explore the molecular mechanisms through which different levels of GAA affect chicken muscle development by influencing miRNA expression, to lay a theoretical foundation for the identification of key functional small RNAs related to poultry muscle development, and to provide new insights into the regulatory mechanisms of GAA on muscle development and meat quality in broilers. It provides a new theoretical basis for using GAA as a feed additive to improve feed performance. Small RNA sequencing technology was utilized to obtain the expression profiles of miRNA in the broiler pectoral muscle fed with different levels of GAA (0 g/kg, 1.2 g/kg and 3.6 g/kg). An analysis of differentially expressed miRNAs revealed 90 such miRNAs in the three combination comparisons, with gga-miR-130b-5p exhibiting significant differences across all three combinations. Furthermore, three of the differentially expressed miRNAs were performed by RT-qPCR verification, yielding results consistent with those obtained from small RNA sequencing. Target gene prediction, as well as the GO and KEGG enrichment analysis of differentially expressed miRNAs, indicated their involvement in muscle cell differentiation and other processes, particularly those associated with the MAPK signaling pathway. This study has, thus, provided valuable insights and resources for the further exploration of the miRNA molecular mechanism underlying the influence of guanidine acetic acid on broiler muscle development. Combined with previous studies and small RNA sequencing, adding 1.2 g/kg GAA to the diet can better promote the muscle development of broilers.

**Keywords:** small RNA sequencing; guanidinoacetic acid; broiler; muscle development

## 1. Introduction

For the poultry industry, the expansion and development of chicken myofibers is economically significant. The pectoral muscles, as a crucial skeletal muscle in poultry, play a vital role in meat production. Therefore, a focused research effort has been made in the field of genetics and the breeding of poultry in order to comprehend the molecular principles behind the growth and development of muscular tissues, including the pectoral muscle.

Guanidinoacetic acid (GAA), also referred to as guanidine acetate or N-imidylglycine, is classified as an amino acid derivative. It is synthesized from glycine and arginine in the kidneys and, subsequently, forms creatine in the liver through the action of methyltransferase guanidineacetic acid [1,2]. GAA serves as a precursor to creatine and its phosphorylated derivative, creatine phosphate [3]. Because it acts as a temporary energy source, creatine is essential to the body's energy metabolism. The body can regulate ATP supply through the phosphogenic system, which includes free creatine and creatine phosphate, to support the energy requirements for animal growth and development [4]. Guanidineacetic acid, as its precursor, has great stability and acceptable bioavailability, making it a feasible substitute for creatine, considering its expensive cost and instability [5].

Muscle energy supply is important in fast-growing broiler chickens. Muscles contain adenosine-tri-phosphate (ATP), which is a source of energy. PCREA and CREA kinase (CK), located in skeletal muscle, maintain the homeostasis of ATP and adenosine-di-phosphate (ADP) by regenerating ATP from ADP in both the cytosol and mitochondria [6,7]. Vivo experiments proved that when the mouse body lacks Cr, GAA can provide energy for the body under the catalysis of CK. It also shows that CK can use GAA and go through the phosphorylation pathway to fight against energy damage [8,9]. And the main reason why Cr contributes to the synthesis of muscle tissue protein, improves muscle energy reserves and muscle strength is that it is synthesized with the precursor substance GAA. According to Ahmadipour et al. [10], adding 0.5–2.0 g/kg of GAA to broiler diets can considerably increase weight growth and decrease feed-to-gain ratios while having no discernible impact on feed intake. RINGEL et al. [5] found that the ATP/ADP ratio inbroiler breast meat increased linearly with an increase in the GAA supplemental level at 0–0.6 g/kg. Amiri et al. [11] found that the dietary addition of 1.2 g/kg GAA with a reduced concentration of crude protein in broilers can increase body weight and average daily gain, and a reduced FCR.

MicroRNAs, also known as miRNAs, are a group of non-coding RNAs that exhibit a high degree of evolutionary conservation. They are endogenous, single-stranded, small RNA molecules typically consisting of 18 to 25 nucleotides [12]. MiRNAs can exert post-transcriptional control of genes by degrading or inhibiting mRNA production through precise base pairing with target mRNA [13]. These molecules play a crucial role in various processes related to muscle development, including proliferation, differentiation, and apoptosis [14–16]. For instance, miR-1 facilitates muscle generation by targeting HDAC4, while the overexpression of miR-133 in myoblasts suppresses SRF-mediated promotion of muscle cell proliferation [17,18]; miR-206 and miR-486 induce myoblast differentiation and down-regulate Pax7 by directly targeting their 3′UTR [19,20]; miR-2954 inhibits the proliferation of chicken myoblasts and promotes their differentiation through the YY1 gene [21]; miR-16 promotes myoblast apoptosis by down-regulating the expression activities of FOXO1 and BCL2 [22]; miR-7 regulates myoblast proliferation, differentiation, and apoptosis by targeting the expression of the KLF4 gene [23].

The impact of GAA on muscle development has been studied extensively, although the molecular processes behind this effect are still mostly unknown. There are still many unanswered questions regarding miRNA control in the development of muscles. This study focused on Cobb broiler chickens as the subject of investigation and utilized small RNA sequencing technology to investigate the miRNA regulatory mechanism of GAA's effect on chicken muscle development. We have investigated the molecular processes by which varying GAA concentrations impact the expression of miRNA in chicken muscle development. Therefore, our research aims to establish a theoretical basis for identifying key functional small RNAs related to poultry muscle development and provide new insights into the regulatory mechanism of GAA on muscle development and meat quality in broilers, offering a new theoretical foundation for using GAA as a feed additive to improve performance.

## 2. Materials and Methods

### 2.1. Ethical Statement

All the operations and experiment procedures were approved by the Life Sciences Ethics Committee of Yunnan Agricultural University (Approval ID: 202203094), and complied with the national standard of Laboratory Animal-Guideline for the ethical review of animal welfare [24] and the Guide for the Care and Use of Laboratory Animals: Eighth Edition.

### 2.2. Feeding Management and Sample Collection

Ninety-one-day-old Cobb broiler hens with similar birth weight were purchased from Hunan Shuncheng Industrial Co., Ltd., (Shuncheng, China) and randomly divided into 3 groups with 5 replicates per group and 6 chickens per replicate. And they were reared in

a tiered cage under the standard feeding environment conditions including temperature, humidity, and ventilation. The basal diet consisted of corn–soybean meal pellet feed, and the control group was exclusively fed this diet. The experimental groups, namely the normal GAA group and high GAA group, were provided with the basal diet supplemented with 1.2 g/kg and 3.6 g/kg of guanidine acetic acid, respectively, until reaching 42 days of age for slaughter and sampling. Subsequently, the left pectoral muscle samples from three chickens in each group were collected, rapidly frozen in liquid nitrogen, and stored at $-80\ ^\circ\mathrm{C}$ for subsequent analysis.

### 2.3. RNA Isolation, Construction and Sequencing of Small RNA Libraries

Total RNA was extracted using Trizol reagent. The extent of RNA degradation and potential contamination was assessed through agarose gel electrophoresis, while the purity of RNA was ascertained using Nanodrop. The concentration of the RNA was precisely quantified using Qubit, and the integrity of RNA was accurately assessed using Agilent 2100. A small RNA library was created from pectoral muscle tissue using a Small RNA Sample Prep Kit once the samples had been qualified. Using whole RNA as the starting material, the distinct 3′ and 5′ end structures of small RNA—which are distinguished by a full phosphate group at the 5′ end and a hydroxyl group at the 3′ end—were used. The two ends of small RNA were directly ligated, followed by reverse transcription to synthesize cDNA. The target DNA fragment was separated using PAGE gel electrophoresis with the aid of subsequent PCR amplification, and the cDNA library was recovered by gel excision. Upon library construction, preliminary quantification was performed using Qubit 2.0, and the library was diluted to 1 ng/μL. The insert size of the library was determined using Agilent 2100. The effective concentration of the library was accurately measured using RT-qPCR, with a requirement of greater than 2 nM. Qualified libraries were subsequently subjected to sequencing using Illumina SE50.

### 2.4. Quality Control of Small RNA Sequencing

After processing the original sequencing reads, the length distribution of the small RNA reads was calculated, and the sequencing quality was evaluated. Sequences with an N content exceeding 10%, low-quality sequences, those with 5′ splices, lacking 3′ splices or inserted fragments, and sequences containing ployA/T/G/C were excluded. For the ensuing analysis, clean readings in the 18–35 nt range were used. These length-screened clean reads were aligned to the chicken genome using bowtie and annotated for rRNA, tRNA, snRNA, snoRNA, repeats, and others. The remaining sequences were then compared with the chicken miRBase database to identify known miRNAs, while unannotated sequences were compared with chicken genome sequences to detect potential novel miRNAs. Utilizing miREvo and miRDeep2 for the prediction of a hairpin structure and folding energy, only sequences exhibiting a stem-loop hairpin structure were taken into consideration as possible candidates for novel miRNAs.

### 2.5. Differential Expression Analysis and RT-qPCR Validation

The levels of expression of both known and novel miRNAs in each sample were tallied, and these levels were then normalized using transcripts per million (TPM). The normalized expression value was determined as follows: (read count $\times$ 1,000,000)/total miRNA read count in the library. A screening criterion of $p$-value $< 0.05$ and $|\log2(\text{fold change})| > 1$ was used to identify differentially expressed miRNAs after the samples were analyzed using the DESeq2 program based on a negative binomial distribution. After selecting differentially expressed miRNAs at random, the reverse transcription quantitative polymerase chain reaction (RT-qPCR) was used to confirm the expression levels of the selected miRNAs. Total RNA was extracted from chest muscle tissue samples using TRIzol reagent (Takara, Dalian, China), and reverse transcription was carried out using the primerScript™RT reagent Kit with gDNA Eraser (Takara). The RT-qPCR analysis of miRNAs was conducted using the TB Green®Premix Ex TaqTMII kit (Takara). The primer information for these

miRNAs was provided in Table 1. The PCR amplification process involved an initial step at 95 °C for 5 min, followed by 40 cycles of 95 °C for 30 s, 61 °C for 30 s, and 72 °C for 30 s, and the last step was conducted at 72 °C for 10 min. PCR reactions were performed using the CFX Connect Real-Time System (BIO-RAD, Hercules, CA, USA). All reactions were repeated three times, and the relative expression was determined using the $2^{-\Delta\Delta Ct}$ method. The U6 small nuclear RNA served as the internal control for miRNA. The statistical significance of the expression levels was evaluated using the *t*-test (unpaired, two-tailed), with a significance level set at $p \leq 0.05$.

**Table 1.** The primer sequences of the stem-loop RT-qPCR experiments.

| miRNA Name | Primer | Primer Sequence |
|---|---|---|
| gga-miR-130b-5p | Loop | GTTGGCTCTGGTGCAGGGTCCGAGGTATTCGCACCAGAGCCAACAGTAGT |
|  | F | TGTGTTTTCCTCTTTCCCTGTTG |
|  | R | GTGCAGGGTCCGAGGT |
| gga-miR-1a-3p | Loop | GTTGGCTCTGGTGCAGGGTCCGAGGTATTCGCACCAGAGCCAACTACATA |
|  | F | TGTTGTGGGTGGAATGTAAAGAAG |
|  | R | GTGCAGGGTCCGAGGT |
| gga-miR-19a-3p | Loop | GTTGGCTCTGGTGCAGGGTCCGAGGTATTCGCACCAGAGCCAACTCAGTT |
|  | F | GGTTTTTTTTTTGTGCAAATCTATGCAA |
|  | R | GTGCAGGGTCCGAGGT |

### 2.6. Differentially Expressed miRNA Target Gene Prediction, GO and KEGG Enrichment Analysis

The miRNA target genes were forecasted by using two software tools, miRanda 3.3a and RNAhybrid v2.0, and the common set of predictions was chosen. Subsequently, the Gene Ontology (GO) enrichment analysis was utilized to elucidate the functions of the target genes. Additionally, pathway analysis was performed using the KEGG database to identify the significant pathways associated with the target genes, with a threshold of *p*-value < 0.05 indicating significant enrichment.

## 3. Results

### 3.1. Quality Control of Small RNA Sequencing Data

Following small RNA sequencing, the control group yielded 39917681 original sequences, while the normal GAA and high GAA groups produced 46448889 and 45750599 original sequences, respectively (Table A1). After the removal of sequences containing over 10% N content, low-quality sequences, those with 5′ splices, lacking 3′ splices or insertions, and those containing ployA/T/G/C, the clean reads obtained were 38330614, 44975675, and 44986930, respectively. The Q20 and Q30 quality tests both passed, with the required error rates being less than 0.01% (Table A2). These results show that the sequencing results are appropriate for additional data processing. Subsequently, small RNAs within the 18–35 nt range were selected from the obtained clean reads for further analysis (Table A3). The sequencing revealed that small RNAs were predominantly distributed within the 20–24 nt range (Figure 1), indicative of typical Dicer enzyme cleavage products. Notably, the peak length of small RNAs was concentrated at 22 nt, consistent with the distribution of animal miRNA length sequencing. The main text should cite all the figures and tables as Figure 1, Table 1, etc.

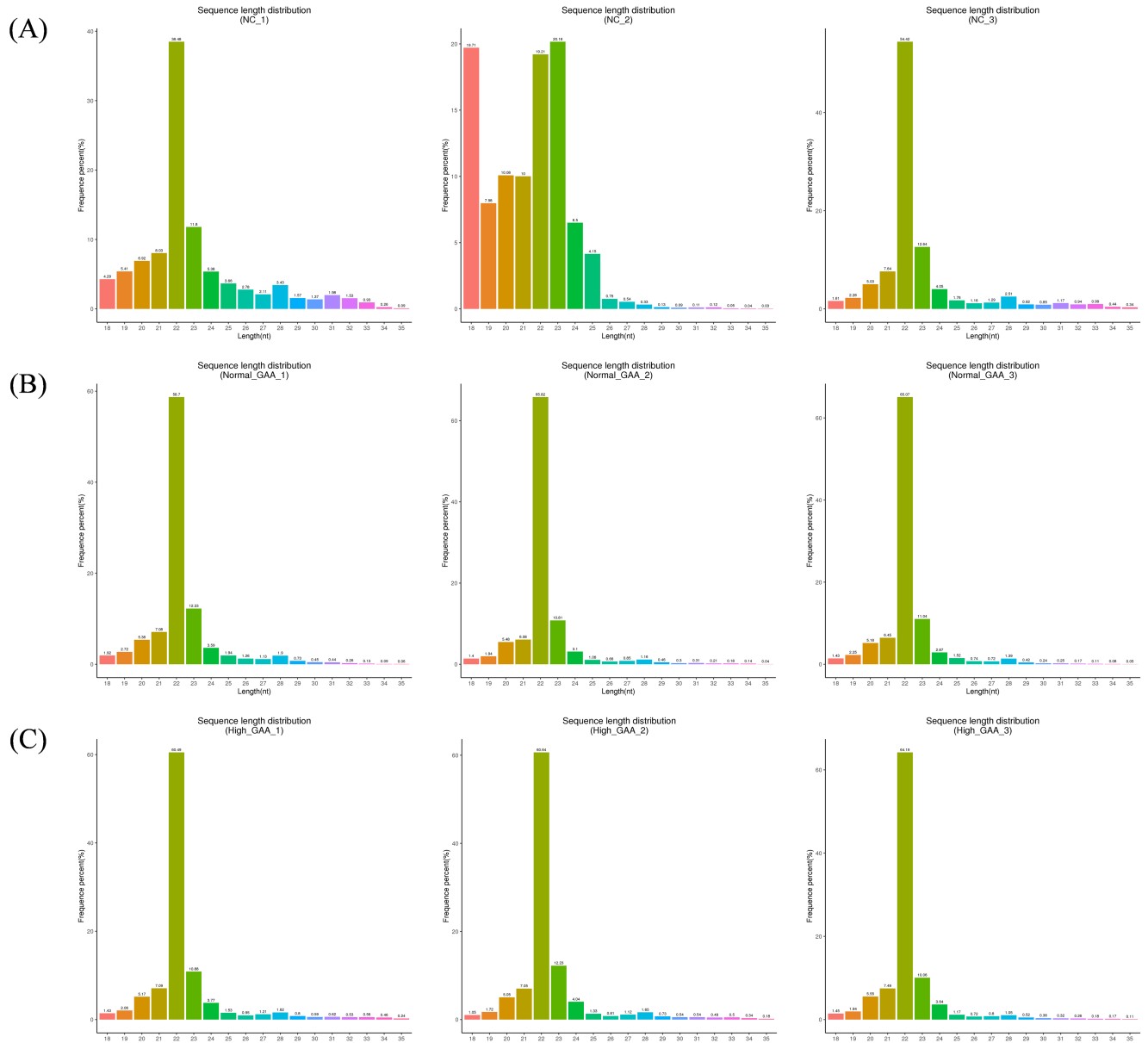

**Figure 1.** The length distribution statistics of total sRNA fragments were obtained. "Control" represents the broilers fed the basal diet; "Normal GAA" represents the broilers fed the diet with 1.2 g GAA; "High GAA" represents the broilers fed the diet with 3.6 g GAA; (**A**) is the total sRNA fragment length distribution obtained from the three samples in the control group; (**B**) is the total sRNA fragment length distribution obtained from the three samples in the Normal GAA group; (**C**) is the total sRNA fragment length distribution obtained from the three samples in the High GAA group.

### 3.2. Genome Comparison and Classification Notes

The small RNA (sRNA) was aligned to the reference genome using bowtie. On average, 90.71%, 95.41%, and 95.90% of the clean reads in the control group, normal GAA group, and high GAA group, respectively, were mapped to the reference sequence. Among these, 67.59%, 75.61%, and 78.06% were aligned to the reference sequence in the same direction, while 23.12%, 19.80%, and 17.85% were aligned in the opposite direction (Table 2). A priority order of known miRNA > rRNA > tRNA > snRNA > snoRNA > repeat > NAT-siRNA > gene > novel miRNA > ta-siRNA was employed for classification in order to guarantee unique annotation for each sRNA (Tables 3 and 4). When compared with the chicken miRBase database, an average of 47.16%, 73.72%, and 75.97% of sequences in

the control group, normal GAA group, and high GAA group were annotated as known miRNAs, with very few types of miRNAs (0.77%, 0.98%, and 0.77%, respectively). Repeated sequences accounted for approximately 0.26%, 0.11%, and 0.09% in the control group, normal GAA group, and high GAA group; additionally, these types of repeated sequences had a proportion of 2.54%, 2.49%, and 2.06%, respectively. Prediction software such as miREvo [25] and mirdeep2 version 2.0.1.2 [26] were utilized to analyze new miRNAs, revealing approximately 0.06%, 0.03%, and 0.06% new miRNAs in the control group, normal GAA group, and high GAA group, with a proportion of new miRNA species of 0.04%, 0.03%, and 0.03%, respectively. Additionally, approximately 14.50%, 13.42%, and 12.51% of sequences in the control group, normal GAA group, and high GAA group were unannotated.

**Table 2.** Comparisons with reference sequences.

| Sample | Total sRNA | Mapped sRNA | +Mapped sRNA | −Mapped sRNA |
|--------|-----------|-------------|--------------|--------------|
| NC_1 | 6,294,787 (100.00%) | 5,835,551 (92.70%) | 3,938,059 (62.56%) | 1,897,492 (30.14%) |
| NC_2 | 8,083,155 (100.00%) | 6,821,769 (84.39%) | 5,655,641 (69.97%) | 1,166,128 (14.43%) |
| NC_3 | 14,262,690 (100.00%) | 135,553,165 (95.03%) | 10,017,664 (70.24%) | 3,535,501 (24.79%) |
| Normal_GAA_1 | 13,645,431 (100.00%) | 12,975,568 (95.09%) | 9,757,778 (71.51%) | 3,217,790 (23.58%) |
| Normal_GAA_2 | 13,575,735 (100.00%) | 12,976,213 (95.58%) | 10,679,295 (78.66%) | 2,296,918 (16.92%) |
| Normal_GAA_3 | 13,806,341 (100.00%) | 13,193,544 (95.56%) | 10,582,156 (76.65%) | 2,611,388 (18.91%) |
| High_GAA_1 | 14,263,796 (100.00%) | 13,609,746 (95.41%) | 11,012,433 (77.21%) | 2,597,313 (18.21%) |
| High_GAA_2 | 14,496,158 (100.00%) | 13,803,787 (95.22%) | 11,036,143 (76.13%) | 2,767,644 (19.09%) |
| High_GAA_3 | 14,185,836 (100.00%) | 13,770,873 (97.07%) | 11,467,097 (80.83%) | 2,303,776 (16.24%) |

Notes: Sample is the sample id; Total sRNA: clean reads of each sample obtained after length screening; Mapped sRNA: the number and percentage of clean reads mapped to the reference sequence in the sample; +Mapped sRNA: the number and percentage of reads in the clean reads of the sample that are mapped to the same direction of the reference sequence; −Mapped sRNA: the number and percentage of reads in the clean reads of the sample that are mapped to the chain in the opposite direction of the reference sequence.

### 3.3. Screening and RT-qPCR Validation of Differentially Expressed miRNAs

Each sample's levels of known and novel miRNAs were measured, and TPM was used to standardize the expression levels of each miRNA. DESeq2 was used to perform pairwise comparisons of samples based on a negative binomial distribution (Figure 2). Upon comparison and analysis, 16 miRNAs exhibited significant differences in expression between the normal GAA group and the control group, with 4 miRNAs being up-regulated and 16 miRNAs being down-regulated (Figure 2A). Additionally, 60 miRNAs showed significant expression differences between the high GAA group and the control group, with 12 miRNAs being up-regulated and 48 miRNAs being down-regulated (Figure 2B). Additionally, compared to the high GAA group and the normal GAA group, 49 miRNAs showed significant expression variations; these included 21 miRNAs with up-regulated expression and 28 miRNAs with down-regulated expression (Figure 2C). Notably, 8 miRNAs were the common differentially expressed miRNAs for normal GAA group vs. control group and high GAA group vs. control group, while 3 miRNAs exhibited differential expression between the normal GAA group and the control group, and between the high GAA group and the normal GAA group. Furthermore, a difference in the expression of 22 miRNAs was observed between the groups with high GAA and normal GAA, as well as between the high GAA group and the normal GAA group. Notably, gga-miR-130b-5p

was identified as the differentially expressed miRNA in the comparison of the three groups (Figure 2E). The differentially expressed miRNAs from nine samples were subjected to hierarchical cluster analysis in order to determine whether GAA had an impact on the expression of miRNA. The clustering results distinctly segregated the 9 samples into three main branches, indicating good consistency among the samples and the rational design of the study (Figure 2D). A number of these differentially expressed miRNAs, including gga-miR-148a-3p and gga-miR-1a-3p, were discovered to be connected to the development of muscle. Furthermore, three of the differentially expressed miRNAs were randomly selected for RT-qPCR verification, and the results were consistent with the small RNA sequencing (Figure 2F), suggesting that the findings of this small RNA sequencing were reproducible and reliable, and could be utilized for further data analysis.

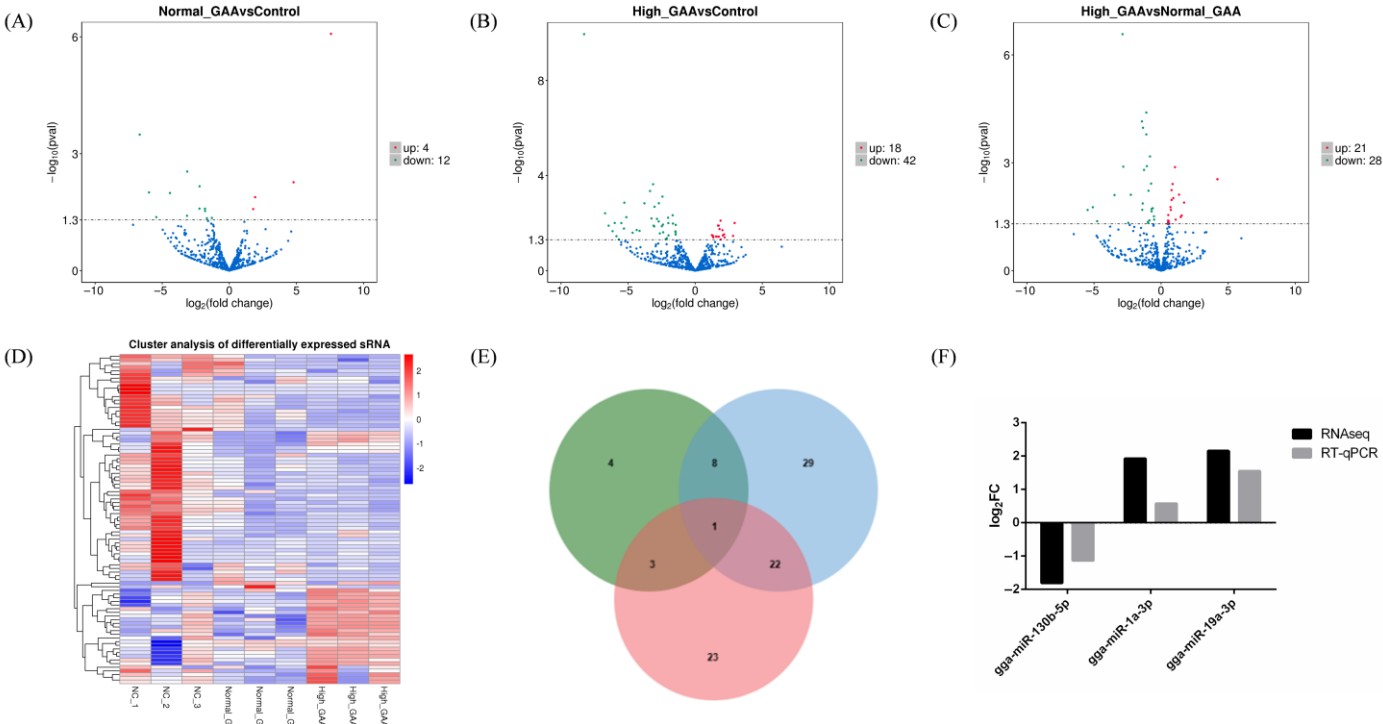

**Figure 2.** (**A**) Differentially expressed miRNA volcano map for Normal GAA group vs. control group in small RNA sequencing analysis; (**B**) Differentially expressed miRNA volcano map for High GAA group vs. control group in small RNA sequencing analysis; (**C**) Differentially expressed miRNA volcano map for High GAA group vs. Normal GAA group in small RNA sequencing analysis; (**D**) Differentially expressed miRNA cluster diagram; (**E**) Wayne diagram; (**F**) Illustration of qPCR confirmation results for three selected differentially expressed miRNAs.

*3.4. Prediction and Bioinformatics Analysis of Differentially Expressed miRNA Target Genes*

Following the identification of differentially expressed miRNAs among the groups, the target genes of these miRNAs were screened based on their correspondence. A total of 636 target genes were identified from 16 miRNAs with significant expression differences between the normal GAA group and the control group. Additionally, 2374 target genes were identified from 60 miRNAs with significant expression differences between the high GAA group and the control group, and 1669 target genes were identified from 49 miRNAs with significant expression differences between the high GAA group and the normal GAA group. Afterwards, Gene Ontology (GO) and Kyoto Encyclopedia of Genes and Genomes (KEGG) analyses were used to enrich the target gene sets of the differentially expressed miRNAs in each group. The GO analysis revealed that, compared to the control group, the target genes in the normal GAA group were primarily enriched in cellular

composition, single-organism processes, somatic processes, localization, propagation, and establishment of localization (Figure 3A), while the target genes in the high GAA group were mainly enriched in catalytic activity, cellular components, and transferase activity compared to the control group (Figure 3B). Furthermore, compared to the normal GAA group, the target genes in the high GAA group were mainly enriched in protein binding and transferase activity (Figure 3C). According to the KEGG pathway enrichment analysis, the target genes in the normal GAA group were primarily enriched in metabolic pathways, endocytosis, MAPK signaling pathways, and RNA transport as compared to the control group (Figure 4A). Similarly, compared to the control group, the target genes in the high GAA group were mainly enriched in metabolic pathways, MAPK signaling pathways, endocytosis, and adhesion plaques (Figure 4B). Finally, compared to the normal GAA group, the target genes were primarily abundant in metabolic pathways, MAPK signaling pathways, adhesion plaques, and endocytosis (Figure 4C).

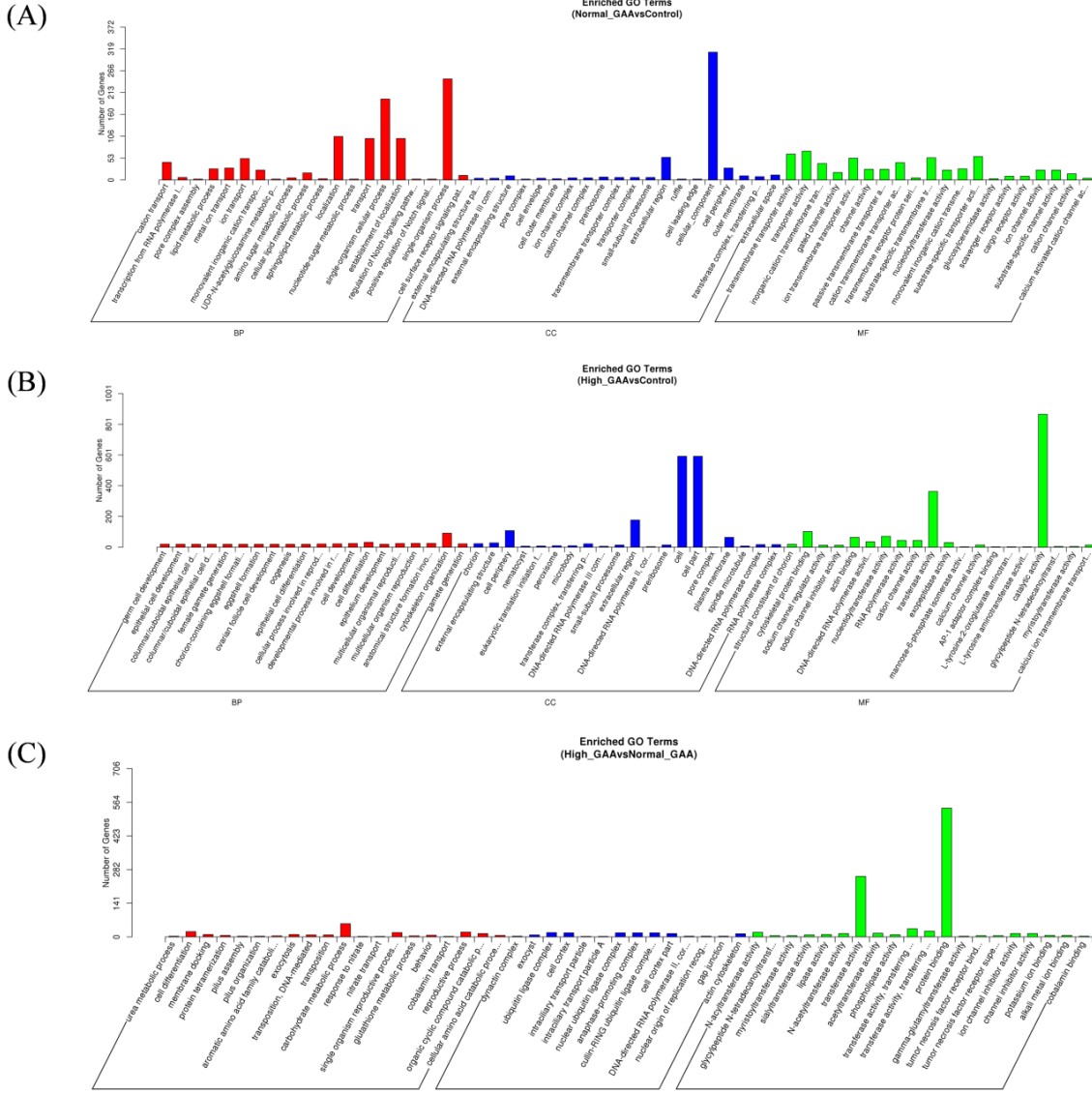

**Figure 3.** GO enrichment column of candidate target genes. (**A**) GO functional enrichment histogram of candidate target genes for Normal GAA group vs. control group in small RNA sequencing analysis; (**B**) GO functional enrichment histogram of candidate target genes for High GAA group vs. control group in small RNA sequencing analysis; (**C**) GO functional enrichment histogram of candidate target genes for High GAA group vs. Normal GAA group in small RNA sequencing analysis.

**Table 3.** The number of sRNA types above.

| Types | NC_1 | NC_2 | NC_3 | Normal_GAA_1 | Normal_GAA_2 | Normal_GAA_3 | High_GAA_1 | High_GAA_2 | High_GAA_3 |
|---|---|---|---|---|---|---|---|---|---|
| total | 5,835,551 (100.00%) | 6,821,769 (100.00%) | 13,553,165 (100.00%) | 12,975,568 (100.00%) | 12,976,213 (100.00%) | 13,193,544 (100.00%) | 13,609,746 (100.00%) | 13,803,787 (100.00%) | 13,770,873 (100.00%) |
| known_miRNA | 2,520,080 (43.18%) | 2,099,008 (30.77%) | 9,152,611 (67.53%) | 8,735,596 (67.32%) | 10,154,260 (78.25%) | 9,971,723 (75.58%) | 9,975,678 (73.30%) | 10,387,778 (75.25%) | 10,926,503 (79.35%) |
| rRNA | 1,425,155 (24.42%) | 1,898,055 (27.82%) | 1,204,404 (8.89%) | 1,369,240 (10.55%) | 490,065 (3.78%) | 836,735 (6.34%) | 961,446 (7.06%) | 717,176 (5.20%) | 821,914 (5.97%) |
| tRNA | 144,481 (2.48%) | 73,725 (1.08%) | 257,837 (1.90%) | 48,019 (0.37%) | 104,579 (0.81%) | 33,326 (0.25%) | 179,490 (1.32%) | 141,007 (1.02%) | 56,612 (0.41%) |
| snRNA | 1633 (0.03%) | 13,109 (0.19%) | 3240 (0.02%) | 2433 (0.02%) | 2091 (0.02%) | 1459 (0.01%) | 4170 (0.03%) | 3218 (0.02%) | 2712 (0.02%) |
| snoRNA | 20,339 (0.35%) | 78,328 (1.15%) | 49,895 (0.37%) | 27,276 (0.21%) | 30,450 (0.23%) | 18,055 (0.14%) | 52,642 (0.39%) | 51,549 (0.37%) | 33,589 (0.24%) |
| repeat | 26,939 (0.46%) | 12,204 (0.18%) | 17,813 (0.13%) | 18,937 (0.15%) | 13,861 (0.11%) | 8890 (0.07%) | 13,933 (0.10%) | 12,478 (0.09%) | 9691 (0.07%) |
| novel_miRNA | 2385 (0.04%) | 5318 (0.08%) | 7863 (0.06%) | 3611 (0.03%) | 5041 (0.04%) | 3508 (0.03%) | 6634 (0.05%) | 10,407 (0.08%) | 7171 (0.05%) |
| exon | 592,596 (10.15%) | 220,250 (3.23%) | 441,720 (3.26%) | 543,749 (4.19%) | 232,465 (1.79%) | 251,753 (1.91%) | 297,656 (2.19%) | 314,672 (2.28%) | 216,897 (1.58%) |
| intron | 196,410 (3.37%) | 1,492,176 (21.87%) | 471,223 (3.48%) | 419,346 (3.23%) | 281,334 (2.17%) | 285,090 (2.16%) | 348,324 (2.56%) | 243,553 (1.76%) | 235,124 (1.71%) |
| other | 905,533 (15.52%) | 929,596 (13.63%) | 1,946,559 (14.36%) | 1,807,361 (13.93%) | 1,662,067 (12.81%) | 1,783,005 (13.51%) | 1,769,773 (13.00%) | 1,921,949 (13.92%) | 1,460,660 (10.61%) |

Notes: total: refers to the number of sRNAs of each sample compared to the reference sequence; known_miRNA: refers to the number and proportion of sRNAs that are compared to known miRNAs in each sample; rRNA/tRNA/snRNA/snoRNA: refers to the number and proportion of sRNAs that are compared to rRNA/tRNA/snRNA/snoRNA in each sample; repeat: refers to the number and proportion of sRNA that each sample is compared to the repeat; novel_miRNA: refers to the number and proportion of sRNAs that are compared to new miRNAs in each sample; exon/intron: refers to the number and proportion of sRNAs compared to exon/intron for each sample; other: refers to the number and proportion of each sample to the reference sequence, but not to known_miRNA, ncRNA, repeat, novel_miRNA, and sRNA in gene exon and intron region.

**Table 4.** The types of sRNA compared above.

| Types | NC_1 | NC_2 | NC_3 | Normal_GAA_1 | Normal_GAA_2 | Normal_GAA_3 | High_GAA_1 | High_GAA_2 | High_GAA_3 |
|---|---|---|---|---|---|---|---|---|---|
| total | 219,401 (100.00%) | 284,669 (100.00%) | 278,624 (100.00%) | 239,364 (100.00%) | 284,463 (100.00%) | 186,060 (100.00%) | 358,971 (100.00%) | 342,354 (100.00%) | 268,206 (100.00%) |
| known_miRNA | 1453 (0.66%) | 2362 (0.83%) | 2304 (0.83%) | 1931 (0.81%) | 2068 (0.73%) | 2050 (1.10%) | 2443 (0.68%) | 2491 (0.73%) | 2411 (0.90%) |
| rRNA | 22,849 (10.41%) | 46,812 (16.44%) | 37,576 (13.49%) | 33,722 (14.09%) | 32,757 (11.52%) | 31,463 (16.91%) | 48,641 (13.55%) | 44,012 (12.86%) | 43,990 (16.40%) |
| tRNA | 1895 (0.86%) | 4136 (1.45%) | 3312 (1.19%) | 2534 (1.06%) | 3012 (1.06%) | 2,308 (1.24%) | 3841 (1.07%) | 4032 (1.18%) | 3053 (1.14%) |
| snRNA | 318 (0.14%) | 1434 (0.50%) | 605 (0.22%) | 482 (0.20%) | 560 (0.20%) | 345 (0.19%) | 910 (0.25%) | 724 (0.21%) | 686 (0.26%) |
| snoRNA | 1240 (0.57%) | 4672 (1.64%) | 2114 (0.76%) | 1612 (0.67%) | 1909 (0.67%) | 1343 (0.72%) | 2803 (0.78%) | 2458 (0.72%) | 2240 (0.84%) |
| repeat | 7059 (3.22%) | 5532 (1.94%) | 6842 (2.46%) | 6287 (2.63%) | 7356 (2.59%) | 4184 (2.25%) | 7813 (2.18%) | 6900 (2.02%) | 5275 (1.97%) |
| novel_miRNA | 75 (0.03%) | 123 (0.04%) | 98 (0.04%) | 82 (0.03%) | 90 (0.03%) | 78 (0.04%) | 114 (0.03%) | 98 (0.03%) | 95 (0.04%) |
| exon | 71,761 (32.71%) | 85,408 (30.00%) | 86,346 (30.99%) | 74,266 (31.03%) | 79,096 (27.81%) | 57,626 (30.97%) | 111,909 (31.17%) | 101,894 (29.76%) | 84,741 (31.60%) |
| intron | 42,222 (19.24%) | 35,702 (12.54%) | 46,795 (16.80%) | 38,953 (16.27%) | 62,547 (21.99%) | 24,670 (13.26%) | 63,943 (17.81%) | 59,326 (17.33%) | 37,840 (14.11%) |
| other | 70,529 (32.15%) | 98,488 (34.60%) | 92,632 (33.25%) | 79,495 (33.21%) | 95,068 (33.42%) | 61,993 (33.32%) | 116,554 (32.47%) | 120,419 (35.17%) | 87,875 (32.76%) |

Notes: total: refers to the sRNA type of each sample compared to the reference sequence; known_miRNA: refers to the type and proportion of sRNA that is compared to known miRNA in each sample; rRNA/tRNA/snRNA/snoRNA: refers to the type and proportion of sRNA that is compared to rRNA/tRNA/snRNA/snoRNA in each sample; repeat: refers to the type and proportion of sRNA that each sample is compared to the repeat; novel_miRNA: refers to the types and proportion of sRNAs that are compared to new miRNAs in each sample; exon/intron: refers to the types and proportion of sRNAs compared to exon/intron of each sample; other: refers to the types and proportion of SRnas in the exon and intron regions of each sample compared to the reference sequence but not compared to known_miRNA, ncRNA, repeat, novel_miRNA, TAS, and gene exon and intron region.

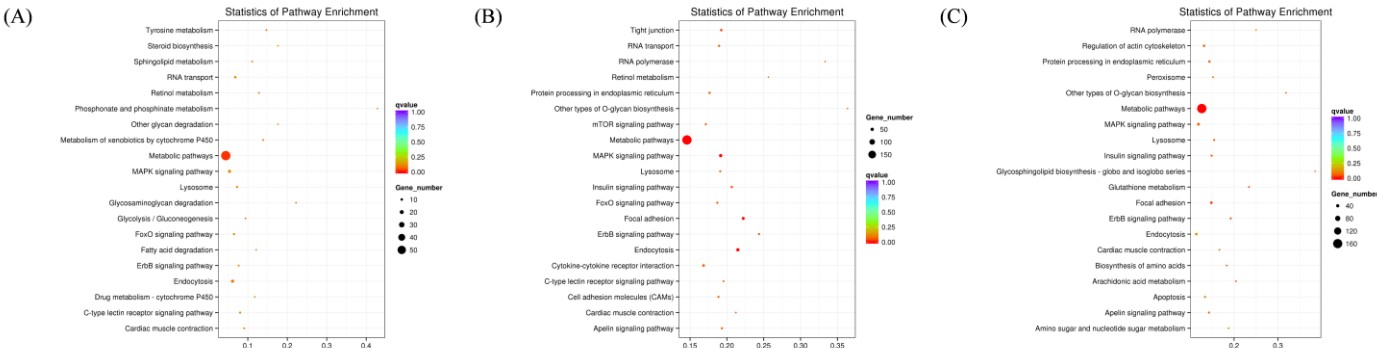

**Figure 4.** KEGG enrichment scatter plot of candidate target genes. (**A**) KEGG enrichment scatter plot of candidate target genes for Normal GAA group vs. control group in small RNA sequencing analysis; (**B**) KEGG enrichment scatter plot of candidate target genes for High GAA group vs. control group in small RNA sequencing analysis; (**C**) GO functional enrichment histogram of candidate target genes for High GAA group vs. Normal GAA group in small RNA sequencing analysis.

## 4. Discussion

In addition to its great agricultural and economic significance, chickens are excellent models for animal experiments. The assessment of poultry meat yield and economic benefit relies heavily on the growth and development of chicken muscle. Studying the patterns of chicken muscle development is essential for enhancing poultry meat production and gaining insights into muscle-related diseases. Wang et al. demonstrated a notable increase in the diameter of mouse muscle fibers following the administration of GAA [27]. Similarly, Lu et al. reported a significant reduction in muscle fiber density and an increase in the cross-sectional area of pig muscle fibers with the dietary supplementation of GAA [28]. According to Michiels [29] and Lemme [30], broiler body weight and breast meat output could be effectively increased by supplementing feed with GAA. Ringel et al. [31] found that the dietary supplementation of guanidine acetic acid (0.6–1.2 g/kg) significantly improved production performance and increased breast muscle ratio. Exogenous GAA supplementation has been demonstrated in studies to enhance feed consumption and average daily growth in Angus cattle [32,33]. Meanwhile, in sheep, performance, carcass characteristics and meat nutritional content have been reported to improve in association with GAA [34,35]. Unlike mammals, environmental stress is more common in broiler production. Several studies have shown that adding GAA can improve the growth performance of heat-stressed chickens. For example, dietary supplementation with GAA (1.2 g/kg) and the reduction of crude protein concentration (90% of normal crude protein concentration) increased body weight and average daily gain, and decreased FCR in heat-stressed broilers [12]. Our earlier research showed that supplementing with 1.2 g/kg GAA greatly improved broiler performance and muscle growth, while supplementing with 3.6 g/kg GAA did not significantly affect the control group's performance [36]. These results suggest that an appropriate level of GAA supplementation in the diet can effectively enhance the development of pectoral muscle. Conversely, excessive GAA supplementation can also promote pectoral muscle development, but to a lesser extent compared to the appropriate dosage.

MicroRNAs (miRNAs), as a type of small non-coding RNAs, can suppress the expression of target genes by binding to the 3′ untranslated region (UTR) of these genes [37]. A single miRNA has the capacity to regulate multiple genes, while a single gene can be regulated by multiple miRNAs [38]. Previous histological analyses have shown that different concentrations of guanidine acetic acid (GAA) in diets can lead to significant differences in muscle development. To investigate the regulatory mechanism of miRNAs in GAA-mediated muscle development, small RNA sequencing was employed to identify the signaling pathway associated with miRNA target genes. Previous research has established that the involvement of miRNAs in animal muscle development, such as miR-128a, targets IRS1 to modulate myoblast proliferation [39]; miR-199a-5p regulates the

proliferation and differentiation of myogenic cells by inhibiting WNT signaling factors [40], miR-221/222 regulates myogenesis in quail through p27 [41], and miR-148a-3p participates in the proliferation and differentiation of bovine muscle cells by targeting KLF4 [42].

This investigation identified a total of 90 differentially expressed miRNAs through small RNA sequencing. Out of all of them, gga-miR-1a-3p had the highest expression level, and, in a pairwise analysis, only gga-miR-130b-5p showed a significant difference among the three groups. Previous studies have shown that GAA can target Insulin Receptor (Insr) and Eukaryotic Translation Initiation Factor 4E (EIF4E) through miR-1a-3p and miR-133a-3p, respectively, and then activate the AKT/mTOR/S6K signaling pathway, thereby stimulating myoblast differentiation [27]. Additionally, miR-130b-5p has been demonstrated to enhance cardiomyocyte proliferation via MAPK-ERK [43]. It was found that Transforming Growth Factor-β1 (TGF-β1) may be the target gene of miR-133a-3p, and GAA can affect the muscle development of broilers through TGF-β signaling pathway [44]. KEGG pathway enrichment analysis of the target genes of differentially expressed miRNAs revealed a significant enrichment in the MAPK signaling pathway. The MAPK signaling pathway is a conserved mechanism involved in various cellular processes such as cell proliferation, differentiation, apoptosis, and oxidative stress [45]. The MAPK signaling cascade can regulate muscle cell proliferation or differentiation by controlling the expression of myogenic transcription factors [46]. The genes Platelet-derived Growth Factor A (PDGFA), Insulin-like Growth Factor-2 (IGF-2), Platelet-derived Growth Factor D (PDGFD), and HRAS proto-oncogene (HRAS) are targeted by gga-miR-1a-3p and gga-miR-130b-5p, respectively, and are all associated with the MAPK signaling pathway. Liu et al. showed, through in vivo studies, that IGF2-edited pigs showed higher skeletal muscle weight [47]. Studies have shown that the HRAS gene can regulate the differentiation of myoblasts, and the PDGFD gene can regulate the proliferation and differentiation of smooth muscle cells [48–50]. It is reported that the PDGFA gene can regulate the proliferation and differentiation of muscle cells [51]. Furthermore, these genes are also regulated by other differentially expressed miRNAs, including gga-miR-206 and gga-miR-12266-5p. These findings suggest that GAA may modulate the expression of multiple genes in the MAPK signaling pathway through gga-miR-1a-3p, gga-miR-130b-5p, and other differentially expressed miRNAs, thereby influencing muscle development. The above mechanisms for the regulation of guanidinoacetic acid on muscle development are summarized in Figure 5.

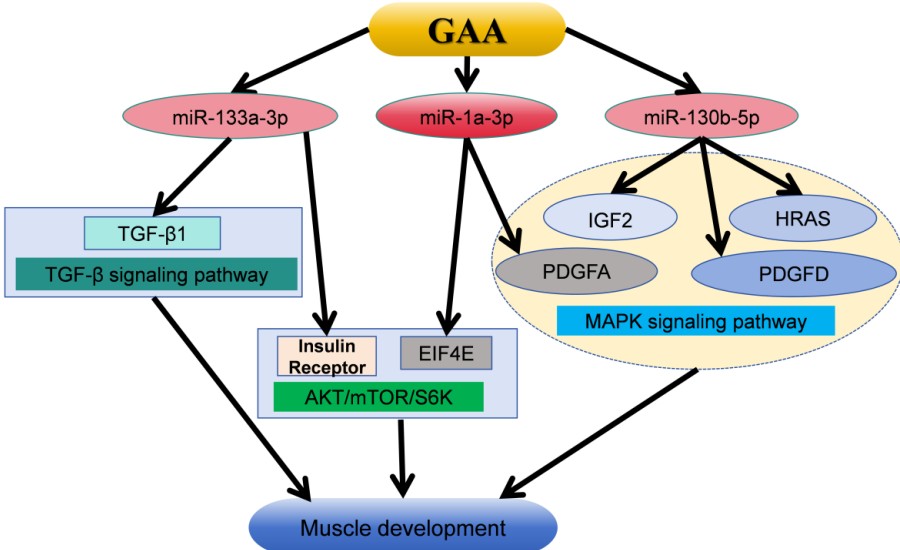

**Figure 5.** The mechanism by which GAA promotes muscle development and growth through mirNA-targeted gene regulatory signaling pathway [27,43,44,47–51]. GAA, guanidinoacetic acid; TGF-β1, Transforming Growth Factor-β1; EIF4E, Eukaryotic Translation Initiation Factor 4E; PDGFA, Platelet-derived Growth Factor A; IGF-2, Insulin-like Growth Factor-2; PDGFD, Platelet-derived Growth Factor D; HRAS, HRAS proto-oncogene.

## 5. Conclusions

In this study, we used small RNA sequencing technology to compare the levels of different GAA in the breast muscle tissue of Cobb broilers, and screened out the differential expression miRNAs related to muscle development, such as gga-miR-1a-3p, gga-miR-130b-5p, etc. Through GO and KEGG enrichment analysis, the target genes of differentially expressed miRNAs were enriched in the MAPK signaling pathway related to muscle development. This suggests, to some extent, that GAA may regulate the signaling pathway through miRNA, and then affect muscle development. However, in order to understand the molecular mechanism of miRNA's role, further verification is needed. Combined with previous studies and small RNA sequencing, adding 1.2 g/kg GAA to the diet can better promote the muscle development of broilers.

**Author Contributions:** M.L. (Mengqian Liu): Writing—original draft, Data curation, Formal Analysis, Software, Visualization. M.L. (Mengyuan Li): Investigation, Methodology, Data curation, Software. J.R.: Investigation, Methodology. J.J.: Conceptualization, Resources. C.G.: Writing—review and editing, Conceptualization, Project administration, Resources. W.C.: Writing—review and editing, Data curation, Funding acquisition, Project administration, Validation. All authors have read and agreed to the published version of the manuscript.

**Funding:** This research was funded by the Natural Science Foundation of Yunnan Province, China (grant No. 202001AU070109); the National Natural Science Foundation of China (grant No. 32202666); the Postdoctoral Orientation training program of Yunnan Province, China; and the open fund of Yunnan Key Laboratory of Animal Nutrition and Feed.

**Institutional Review Board Statement:** All the operations and experiment procedures were approved by the Life Sciences Ethics Committee of Yunnan Agricultural University (Approval ID: 202203094), and complied with the national standard of Laboratory Animal-Guideline for the ethical review of animal welfare and the Guide for the Care and Use of Laboratory Animals: Eighth Edition.

**Informed Consent Statement:** The study did not involve humans.

**Data Availability Statement:** The raw data of small RNA transcriptome presented in the study are deposited in the Sequence Read Archive repository, accession number PRJNA1101858.

**Acknowledgments:** The authors would like to thank all the research assistants and laboratory technicians who contributed to the study. Furthermore, the authors wish to acknowledge the reviewers for their valuable comments and improvements to the manuscript.

**Conflicts of Interest:** The authors declare no conflict of interest.

## Appendix A

**Table A1.** Filtering of sequencing data.

| Sample | Total_Reads | N% > 10% | Low Quality | 5_Adapter_Contamine | 3_Adapter_Null or Insert_Null | with ploya/T/G/C | Clean Reads |
|---|---|---|---|---|---|---|---|
| NC_1 | 9,875,151 (100.00%) | 394 (0.00%) | 18,995 (0.19%) | 105,833 (1.07%) | 312,038 (3.16%) | 13,139 (0.13%) | 9,424,752 (95.44%) |
| NC_2 | 14,441,618 (100.00%) | 365 (0.00%) | 38,177 (0.26%) | 31,738 (0.22%) | 651,214 (4.51%) | 52,777 (0.37%) | 13,667,347 (94.64%) |
| NC_3 | 15,600,912 (100.00%) | 360 (0.00%) | 33,805 (0.22%) | 75,301 (0.48%) | 228,177 (1.46%) | 24,754 (0.16%) | 15,238,515 (97.68%) |
| Normal_GAA_1 | 15,352,341 (100.00%) | 383 (0.00%) | 31,269 (0.20%) | 85,795 (0.56%) | 252,960 (1.65%) | 19,553 (0.13%) | 14,962,381 (97.46%) |
| Normal_GAA_2 | 15,560,784 (100.00%) | 542 (0.00%) | 33,473 (0.22%) | 35,518 (0.23%) | 434,904 (2.79%) | 16,993 (0.11%) | 15,039,354 (96.65%) |
| Normal_GAA_3 | 15,535,764 (100.00%) | 479 (0.00%) | 41,092 (0.26%) | 53,305 (0.34%) | 449,571 (2.89%) | 17,377 (0.11%) | 14,973,940 (96.38%) |
| High_GAA_1 | 15,287,814 (100.00%) | 373 (0.00%) | 38,844 (0.25%) | 39,613 (0.26%) | 167,238 (1.09%) | 27,765 (0.18%) | 15,013,981 (98.21%) |
| High_GAA_2 | 15,166,518 (100.00%) | 381 (0.00%) | 31,076 (0.20%) | 31,112 (0.21%) | 143,446 (0.95%) | 17,570 (0.12%) | 14,942,933 (98.53%) |
| High_GAA_3 | 15,296,267 (100.00%) | 345 (0.00%) | 30,692 (0.20%) | 32,201 (0.21%) | 183,827 (1.20%) | 19,186 (0.13%) | 15,030,016 (98.26%) |

Note: Sample is the sample id; total_reads indicates the number of original sequence data; the number of filtered reads and their proportion to total_reads were N% > 10% because N content exceeded 10%; low quality indicates the number of filtered reads and their proportion in total_reads due to low quality; 5_adapter_contamine the number of filtered reads and their proportion in total_reads because it contains a 5 'joint; 3_adapter_null or insert_null Number of filtered reads and their proportion to total_reads because there is no 3 'joint or no insert_null; with ployA/T/G/C, the number of reads filtered out and their proportion in total_reads because of ployA/T/G/C; the resulting number of clean reads and their proportion in total_reads.

**Table A2.** Summary of the quality of sequencing data output.

| Sample | Reads | Bases | Error Rate | Q20 | Q30 | GC Content |
|---|---|---|---|---|---|---|
| NC_1 | 9,875,151 | 0.494 G | 0.01% | 99.57% | 98.26% | 48.60% |
| NC_2 | 14,441,618 | 0.722 G | 0.01% | 99.37% | 97.71% | 52.39% |
| NC_3 | 15,600,912 | 0.780 G | 0.01% | 99.54% | 98.21% | 46.58% |
| Normal_GAA_1 | 15,352,341 | 0.768 G | 0.01% | 99.55% | 98.30% | 46.04% |
| Normal_GAA_2 | 15,560,784 | 0.778 G | 0.01% | 99.53% | 98.07% | 45.44% |
| Normal_GAA_3 | 15,535,764 | 0.777 G | 0.01% | 99.54% | 98.19% | 45.46% |
| High_GAA_1 | 15,287,814 | 0.764 G | 0.01% | 99.47% | 97.71% | 45.92% |
| High_GAA_2 | 15,166,518 | 0.758 G | 0.01% | 99.53% | 98.27% | 45.39% |
| High_GAA_3 | 15,296,267 | 0.765 G | 0.01% | 99.59% | 98.43% | 45.36% |

Note: Sample is the sample id; Reads is statistical raw sequence data; Bases were the number of sequencing sequences multiplied by the length of sequencing sequences and converted to G; Error rate is the sequencing error rate; Q20 was the percentage of bases with Phred value greater than 20 in the total base; Q30 is the percentage of bases with Phred value greater than 30 in the total base; GC content is the percentage of the total number of bases G and C combined to calculate the total number of bases.

**Table A3.** The number and types of sRNA after length screening.

| Sample | Total Reads | Total Bases (bp) | Uniq Reads | Uniq Bases (bp) |
|---|---|---|---|---|
| NC_1 | 6,294,787 | 144,153,981 | 292,159 | 6,966,123 |
| NC_2 | 8,083,155 | 171,818,221 | 358,265 | 7,735,712 |
| NC_3 | 14,262,690 | 324,483,031 | 358,329 | 8,775,554 |
| Normal_GAA_1 | 13,645,431 | 304,579,166 | 318,234 | 7,546,472 |
| Normal_GAA_2 | 13,575,735 | 301,417,656 | 362,851 | 8,403,654 |
| Normal_GAA_3 | 13,806,341 | 306,305,999 | 245,337 | 5,756,793 |
| High_GAA_1 | 14,263,796 | 320,833,708 | 453,565 | 10,991,480 |
| High_GAA_2 | 14,496,158 | 325,818,359 | 448,438 | 10,970,521 |
| High_GAA_3 | 14,185,836 | 315,122,981 | 342,272 | 8,174,591 |

Note: Sample is the sample id; total_reads is the total number of sRNAs; Total bases (bp) is the total length of the sRNA; Uniq reads is the type of sRNA; Uniq bases (bp) are the total length of each sRNA.

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
