# Peer review of "Analysis of microRNA Expression Profiles in Broiler Muscle Tissues by Feeding Different Levels of Guanidinoacetic Acid"

_cimb, doi:10.3390/cimb46040231_

Round 1
Reviewer 1 Report
Comments and Suggestions for Authors
Notes and recommendations to the authors of the manuscript:
1. The topic of the article falls within the scope of the scientific journal. The manuscript concerns a current problem related to the biochemistry of digestive processes and protein metabolism in broilers. Apart from scientific, the development also has applied value, since the aim of this investigation is to examine the impact of guanidinoacetic acid (GAA) on miRNA expression in broilers muscle tissue and to elucidate the underlying molecular mechanisms of muscle development. It is well known that the growth and development of chicken myofiber hold significant economic im- 28
portance for the poultry industry;
2. The authors have made a detailed literature review and analysis of the development of poultry nutribiochemistry to date and the influence of some nutritional supplements on proteinogenesis and energy metabolism in muscle in broilers. However, more attention could be paid to the creatine phosphokinase reaction, as an additional energy-supplying biochemical mechanism in the muscles, as well as to the role of the amino acid derivative guanidinoacetic acid;
3. The purpose is not quite correctly formulated and does not fully correspond to the title of the manuscript. It shows that the authors will do a theoretical study rather than an experimental study;
4. In the Material and methods section, it is described that experiments were carried out with broiler chickens from one day to 42 days of age. However, it is not clear how many birds are reared in each group and are they equalized in terms of sex and live weight? Also, isn't 3 birds too little to study as a representative sample?
5. In the Results section, figures 1, 2, 3 and 4 need appropriate scaling. I recommend zooming in 10-15% for better visualization;
6. In the Discussion section, the erudition and professionalism of the author's team to analyze and interpret both their own and the world's scientific data on the research problem is evident. I recommend in this section that the authors dig deeper into the bowels of nutritional biochemistry and avian genetics. Unlike ruminants, pigs and birds have a different energy metabolism and a different fate of some amino acid derivatives, including guanidinoacetic acid. Otherwise, I support the authors in their efforts to adjust the appropriate dosage of the dietary supplement;
7. The conclusions drawn at the end of the manuscript do not accurately reflect the theoretical and practical scientific work performed. The authors only touched on RNA sequencing, but not a word about useful recommendations to the poultry industry on appropriate dosages for nutritional supplements. The scientific-applied value of any fundamental scientific development should not be underestimated.
Comments on the Quality of English LanguageNotes regarding the English language of the manuscript:
The manuscript is written in relatively good and scholarly English. However, I recommend final editing by a professional English-speaking editor
Reviewer 2 Report
Comments and Suggestions for Authors
1. Some grammars should be checked and revised. Here are some examples in the abstract.
Ex. line 15, fed with different levels….
Ex. line 20, of the differentially expressed miRNAs….
Ex. line 20-21, indicated their involvements in muscle cell differentiation and other processes, particularly those associated with the MAPK signaling pathway.
2. In Materials and methods, line 74, what is Kebao broilers? A local strain of broiler chickens?
3. Why birds were raised to age of 42 days instead of 35 days? Where are the growth performance results? Where are the breast muscle development results; at least the relative muscle weight should be presented ?
4. When birds were sampled ? at age of 42 days?
5. Where are the studies to confirm the results of target gene predictions, GO and KEGG enrichment analysis?
6. What the markers for muscle development, such as differentiation marker MyoD, myogenin, myosin heavy chain, commitment marker PX3/7, morphological markers, even the final outcome, the relative muscle weight. Without those markers, the study is only a dataset.
Comments on the Quality of English LanguageModerate editing of English language required
Round 2
Reviewer 2 Report
Comments and Suggestions for Authors
In the discussion, a figure to illustrate the differentially expressed miRNAs with the annotated pathways of GO and KEGG to link with GAA in muscle development is suggested to added in. Some of the references can be cited in the figure to avoid a descript study.
Comments on the Quality of English LanguageModerate editing of English language required
Author Response
To Reviewer 2:
Dear Professor:
We sincerely appreciate your critical and in-depth analysis of our manuscript and are very thankful for your valuable suggestions, which indeed helped us to understand our deficiencies and improve our paper. After carefully studying the review comments, corresponding corrections were made according to each suggestion. We hope this revised version will be bet with your approval. The revised parts of the manuscript have been highlighted in red.
- In the discussion, a figure to illustrate the differentially expressed miRNAs with the annotated pathways of GO and KEGG to link with GAA in muscle development is suggested to added in. Some of the references can be cited in the figure to avoid a descript study.
Answer: Thanks for your valuable suggestion. According to your kind comment, we have consulted more relevant documents. Previous studies have shown that GAA can target Insulin Receptor (Insr) and Eukary-otic Translation Initiation Factor 4E (EIF4E) through miR-1a-3p and miR-133a-3p, re-spectively, and then activate the AKT/mTOR/S6K signaling pathway, thereby stimu-lating myoblast differentiation [1]. Additionally, miR-130b-5p has been demonstrated to enhance cardiomyocyte proliferation via MAPK-ERK [2]. It was found that Trans-forming Growth Factor-β1 (TGF-β1) may be the target gene of miR-133a-3p, and GAA can affect the muscle development of broilers through TGF-β signaling pathway [3]. The genes Platelet-derived Growth Factor A (PDGFA), Insulin-like Growth Factor-2 (IGF-2), Platelet-derived Growth Factor D (PDGFD), and HRAS proto-oncogene (HRAS) are targeted by gga-miR-1a-3p and gga-miR-130b-5p, respectively, and are all associated with the MAPK signaling pathway. Liu et al. showed through in vivo studies that IGF2-edited pigs showed higher skeletal muscle weight [4]. Studies have shown that HRAS gene can regulate the differentiation of myoblasts, and PDGFD gene can regulate the proliferation and differentiation of smooth muscle cell [5-7]. It is reported that PDGFA gene can regulate the proliferation and differentiation of muscle cells [8]. We have added relevant references in the paper and highlighted it in the figure. Among them, only the abstract of the article on the effect of GAA on TGF-β signaling pathway has been found, and we will add the reference immediately after the official publication [9]. Revised parts of the manuscript have been highlighted in red, we mapped the mechanism by which GAA promotes muscle development and growth through miRNA-targeted gene regulatory signaling pathways in the Figure 5. in the revised manuscript. Thanks again.
Figure (Figure 5 in the revised manuscript). The mechanism by which GAA promotes muscle development and growth through mirNA-targeted gene regulatory signaling pathway. GAA, guanidinoacetic acid; TGF-β1, Transforming Growth Factor-β1; EIF4E, Eukaryotic Translation Initiation Factor 4E; PDGFA, Platelet-derived Growth Factor A; IGF-2, Insulin-like Growth Factor-2; PDGFD, Platelet-derived Growth Factor D; HRAS, HRAS proto-oncogene.
- Comments on the Quality of English Language:Moderate editing of English language required
Answer: Thank you for pointing out this problem, and we are very sorry about this deficiency. According to your valuable suggestion, we have invited experts with expertise in technical English editing to review our manuscript, and also checked the text thoroughly to avoid grammatical mistakes and to improve language structure, ensuring the goals and results of the study are clear to the potential readers. Revised parts of the manuscript have been highlighted in red, please review. Thanks again.
References:
[1] WANG Y, MA J, QIU W, et al. Guanidinoacetic acid regulates myogenic differentiation and muscle growth through miR-133a-3p and miR-1a-3p co-mediated Akt/mTOR/S6K signaling pathway [J]. International Journal of Molecular Sciences, 2018, 19(9): 2837.
[2] FENG K, WU Y, LI J, et al. Critical Role of miR-130b-5p in Cardiomyocyte Proliferation and Cardiac Repair in Mice After Myocardial Infarction [J]. Stem Cells, 2024, 42(1): 29-41.
[3] LIU C, WANG L, CHEN W. Preliminary study on miRNA-133a-3p and hypoxia in rat cardiomyocytes[J]. Journal of Regional Anatomy & Operative Surgery, 2016, 25(11): 784-788.
[4]. LIU X, LIU H, WANG M, et al. Disruption of the ZBED6 binding site in intron 3 of IGF2 by CRISPR/Cas9 leads to enhanced muscle development in Liang Guang Small Spotted pigs [J]. Transgenic research, 2019, 28: 141-50.
[5]. DYKSTRA P B, RANDO T A, SMOLKE C D. Modulating myoblast differentiation with RNA-based controllers [J]. PloS one, 2022, 17(9): e0275298.
[6]. ZHANG Z-B, RUAN C-C, LIN J-R, et al. Perivascular adipose tissue–derived PDGF-D contributes to aortic aneurysm formation during obesity [J]. Diabetes, 2018, 67(8): 1549-60.
[7]. KURASAWA K, ARAI S, OWADA T, et al. Autoantibodies against platelet-derived growth factor receptor alpha in patients with systemic lupus erythematosus [J]. Modern rheumatology, 2010, 20(5): 458-65.
[8]. SARAVANAMUTHU S S, GAO C Y, ZELENKA P S. Notch signaling is required for lateral induction of Jagged1 during FGF-induced lens fiber differentiation [J]. Developmental biology, 2009, 332(1): 166-76.
[9] https://www.frontiersin.org/articles/10.3389/fvets.2024.1384028/abstract

Round 3
Reviewer 2 Report
Comments and Suggestions for Authors
The version is accepted
Comments on the Quality of English LanguageMinor editing of English language required